# Altered Cellular Metabolism Is a Consequence of Loss of the Ataxia-Linked Protein Sacsin

**DOI:** 10.3390/ijms252413242

**Published:** 2024-12-10

**Authors:** Laura Perna, Grace Salsbury, Mohammed Dushti, Christopher J. Smith, Valle Morales, Katiuscia Bianchi, Gabor Czibik, J. Paul Chapple

**Affiliations:** 1William Harvey Research Institute, Faculty of Medicine and Dentistry, Queen Mary University of London, London EC1M 6BQ, UK; l.perna@qmul.ac.uk (L.P.); g.a.salsbury@qmul.ac.uk (G.S.); m.dushti@qmul.ac.uk (M.D.); c.j.smith@qmul.ac.uk (C.J.S.); g.czibik@qmul.ac.uk (G.C.); 2Pharmacology & Toxicology Department, Faculty of Medicine, Kuwait University, Kuwait City P.O. Box 24923, Kuwait; 3Barts Cancer Institute, Faculty of Medicine and Dentistry, Queen Mary University of London, London EC1M 6BQ, UK; v.morales@qmul.ac.uk (V.M.); k.bianchi@qmul.ac.uk (K.B.)

**Keywords:** ARSACS, sacsin, mitochondria, metabolism, aerobic glycolysis

## Abstract

Mitochondrial dysfunction is implicated in the pathogenesis of the neurological condition autosomal recessive spastic ataxia of Charlevoix-Saguenay (ARSACS), yet precisely how the mitochondrial metabolism is affected is unknown. Thus, to better understand changes in the mitochondrial metabolism caused by loss of the sacsin protein (encoded by the SACS gene, which is mutated in ARSACS), we performed mass spectrometry-based tracer analysis, with both glucose- and glutamine-traced carbon. Comparing the metabolite profiles between wild-type and sacsin-knockout cell lines revealed increased reliance on aerobic glycolysis in sacsin-deficient cells, as evidenced by the increase in lactate and reduction of glucose. Moreover, sacsin knockout cells differentiated towards a neuronal phenotype had increased levels of tricarboxylic acid cycle metabolites relative to the controls. We also observed disruption in the glutaminolysis pathway in differentiated and undifferentiated cells in the absence of sacsin. In conclusion, this work demonstrates consequences for cellular metabolism associated with a loss of sacsin, which may be relevant to ARSACS.

## 1. Introduction

Autosomal recessive spastic ataxia of Charlevoix-Saguenay (ARSACS) is a childhood-onset neurological disease with neurodevelopmental and neurodegenerative components, resulting in pyramidal spasticity and cerebellar ataxia. ARSACS is caused by mutations in the SACS gene, which encodes sacsin/DNAJC29, a modular protein with multiple domains involved in protein quality control [1]. These include a N-terminal ubiquitin-like domain, regions of homology to the ATPase domain of Hsp90 and a J-protein domain, which can recruit Hsp70 chaperone activity [2]. The precise role of sacsin remains unclear; loss of function, however, results in a complex cellular phenotype, including mitochondrial dysfunction. Specifically, sacsin knockdown neurons have altered mitochondrial distribution and dynamics [3,4,5], with mitochondria accumulating in the soma and proximal dendrites [4]. Analysis of the mitochondrial function in sacsin knockdown cells and ARSACS patient-derived fibroblasts has revealed reduced basal respiration, impairment of oxidative phosphorylation (OXPHOS) and mitochondrial ATP production, plus increased levels of reactive oxygen species (ROS) [3,5]. In addition to undermining mitochondrial energy generation, a loss of sacsin reduces the recruitment of the mitochondrial fission factor Drp1 to prospective sites of division. This is predicted to impede mitochondrial fission and may explain some of the mitochondrial phenotypes observed [3]. A mitochondrial phenotype is also observed in a *Sacs*^(-/-)^ mouse model, where disorganised dendritic fields in the cerebellum and Purkinje cell death are predominant features [4]. Dynamicity of mitochondria is required to meet cellular metabolic needs [6], a task that extends beyond sheer ATP production to include the provision of metabolites for cell signalling and the biosynthesis of macromolecules [7]. The objective of this study was to better understand the mitochondrial deficit in ARSACS and specifically test the hypothesis that mitochondrial metabolism would be disrupted. To achieve this goal, we performed metabolic tracer analysis (MTA) of both ^13^C isotopic-labelled glucose and glutamine by mass spectrometry. This facilitated comparison of the metabolite profiles between control and sacsin knockout cell lines, including measurement of metabolite production and usage through pathways, to gain mechanistic insights into mitochondrial dysfunction associated with a loss of sacsin.

## 2. Results

### 2.1. Increased Aerobic Glycolysis in Sacsin Knockout Cells

To determine how mitochondrial and cellular metabolism is disrupted by a loss of sacsin, we performed targeted metabolomics comparing wild-type control SH-SY5Y cells with an otherwise isogenic sacsin knockout cell line generated by CRISPR/Cas9 genome editing (Appendix A) [8]. For metabolic labelling, SH-SY5Y cells were incubated for 16 h with media containing either uniformly labelled [U-13C] glucose or [U-13C] glutamine [Appendix A]. In each experiment, the glucose or glutamine were metabolised with labelled carbons entering metabolic pathways, including glycolysis, the TCA (tricarboxylic acid) cycle and glutaminolysis (Figure 1A). Principal component analysis (PCA) was used to explore the MTA data, demonstrating clustering among replicates with clear segregation between the control and sacsin knockout samples. For 13C glucose labelling, PC1 (which accounted for 93.99% of the total variance) separated the controls from knockouts (Appendix A), while for 13C glutamine labelling, the controls and knockout cells were distinguished by PC2 (which accounted for 28.33% of the total variance) (Appendix A).

MTA, in both independent experiments where glucose and glutamine were used as tracers, revealed that sacsin knockout cells have decreased glucose levels and increased lactate production relative to the controls (Figure 1B). This may suggest that, in sacsin knockout cells, glucose was used to generate lactate. This tentative conclusion is reinforced by looking at the lactate isotopomer distribution, where more than 80% of the total lactate derived from three labelled glucose carbons (m + 3), while less than 20% derived from other sources (m + 0) (Figure 1C), independent of the genotype. The elevated ratio of lactate to glucose in sacsin knockout is indicative of increased aerobic glycolysis.

We next tested how the inhibition of lactate dehydrogenase A isoenzyme (by sodium oxamate), which is responsible for lactate production, affects the confluency over time of sacsin knockout cells. In cells cultured for up to 72 h, the lack of difference in cell confluency between sacsin knockouts and controls, cultured with either sodium oxamate or the vehicle, may suggest that, in conditions of nutrient abundance the bioenergetic and biosynthetic pathways are not compromised (Figure 1D). Intriguingly, glucose deprivation seemed to have a greater effect on the cells’ confluency of the controls than the sacsin knockout cells, suggesting more powerful countermeasures to nutrient deprivation by the latter (Figure 1E). Given the decreased intracellular glucose and increased lactate levels in sacsin knockout cells, we assessed the levels of these metabolites in spent cell culture media. Compared to the controls, sacsin knockout cell media contained less glucose without a difference in the extracellular lactate levels, indicating a more efficient glucose utilisation associated with a loss of sacsin (Figure 1F,G). In other words, the higher intracellular production of lactate from a greater glucose uptake was ultimately converted towards energy or biosynthesis rather than excreted as a waste product.

Previous studies have identified changes in metabolism through differentiation [9,10]. Given the importance of glutamate and related metabolites in neurotransmission and their relevance to neurological diseases, we next undertook neuronal differentiation of the isogenic control and sacsin knockout cells (Appendix A). Neuronal differentiation of SH-SY5Y cells was confirmed by the induction of β-III tubulin, a neuron-specific marker in both control and sacsin knockouts (Appendix A). Morphologically, the control and sacsin knockout cell lines exhibited comparable features of neuronal maturity, including neurites and neuronal projections (Appendix A). MTA with uniformly labelled [U-13C] glucose and [U-13C] glutamine was then performed. PCA of differentiated metabolites showed PC1 separation of control and knockout cells for both 13C glucose labelling (which accounted for 81.18% of the total variance) (Appendix A) and 13C glutamine labelling (which accounted for 93.99% of the total variance) (Appendix A).

Neuronal differentiation did not change the pattern of reduced intracellular glucose and increased intracellular lactate levels in sacsin knockout SH-SY5Y cells (Figure 1H). Contrastingly, spent cell culture media from differentiated sacsin knockouts contained more lactate, suggesting higher lactate export in knockout than in wild-type neurons (Figure 1I). This, along with the higher intracellular levels of phosphoenol pyruvate (PEP, the penultimate glycolytic intermediate before pyruvate) and alanine (the transamination product of pyruvate), may suggest a differentiation-dependent inhibition of downstream reactions of glucose catabolism-related metabolites.

### 2.2. Alteration in Levels of TCA Cycle Metabolites in Sacsin Knockout Cells

Measurement of the total levels of TCA cycle metabolites showed an increase in fumarate and malate levels in sacsin knockout cells relative to the controls (Figure 2A). Analysis of the isotopomer distribution showed limited differences in the incorporation of carbons from either glucose or glutamine into TCA cycle metabolites between sacsin knockout and control cells (Figure 2B–E). These data suggest that the TCA cycle can be anaplerosed by either glucose or glutamine irrespective of sacsin function.

The TCA cycle is closely connected to the mitochondrial respiratory chain through oxidation of NADH and FADH_2_, derived from redox reactions of substrate catabolism, acting as electron donors in cellular respiration. An increase, although not statistically significant, can be seen in the total ATP (*p*-value 0.0768) and NADH (*p*-value 0.0563), but not in ADP and NAD^+^ (Figure 2F), levels in sacsin knockout cells relative to the controls. The increase in energy donor metabolites may suggest cells lacking sacsin could be metabolically fast and over-energised. We then tested the protein levels of the OXPHOS components (Appendix A), which revealed no difference between the cell lines (Appendix A).

Differentiated sacsin knockout cells also showed evidence of increased levels of TCA cycle metabolites relative to the controls (Figure 2G), together with an increase of total ATP and NADH, and no difference in total ADP and NAD^+^ (Figure 2H), suggesting an increase in oxidative metabolism. The quantification of OXPHOS components (from densitometry of immunoblots) in differentiated cell lines also revealed no difference between sacsin knockout cells and wild-type controls (Appendix A).

### 2.3. Disruption of Glutamine-Related Pathway in Sacsin Knockout Cells

We also examined glutamine-related metabolites, with the MTA showing that sacsin knockout cells had decreased the total levels of glutamate relative to the controls (Figure 3A). Isotopomer distribution analysis showed glutamate metabolites derived from glutamine anaplerosis in sacsin knockouts (Figure 3B,C). Analysis of spent culture media detected that, when labelled with glucose sacsin, knockout media contained higher total levels of glutamate than spent wild-type cell media (Figure 3D).

Differentiated sacsin knockout cells showed a small but significant decrease in the total glutamine levels, while glutamate was unchanged (Figure 3E). However, as in the undifferentiated cell line, differentiated sacsin knockout cells incorporated more carbon from [U-13C] glutamine for the production of glutamate compared to wild-type cells (Figure 3F,G). To further understand the changes in glutamate-related metabolism in neuronally differentiated sacsin knockout cells, we interrogated the existing transcriptomic data [8]. Transcript levels of several genes encoding enzymes involved in the glutamine-related pathway were significantly altered in sacsin knockouts relative to the controls (Figure 3H). Specifically, genes that were upregulated in the absence of sacsin included ionotropic glutamate receptors-glutamate ionotropic receptor AMPA subunit 2 (GRIA2), GABA A Receptor Subunit Delta (GABRD), the transcripts of solute carrier family member 7A5 (SLC7A5), which transport large amino acids such as glutamine [11], and the N-Terminal Glutamine Amidase 1 (WDYHV1). While the downregulated genes included GAD1, responsible for catalysing the production of GABA from glutamate; ABAT, responsible for the catabolism of GABA into succinic semialdehyde, ionotropic; GABA A receptors (GABA Receptor Subunit Gamma3 (GABRG3), GABA Receptor Subunit Gamma2 (GABRG2) and GABA A Receptor Subunit Beta3 (GABRB3)) and GABA type B Receptor Subunit 2 (GABBR2); metabotropic glutamate receptor (GMR8) and glutamate receptor-interacting protein 1 (GRIP1), which is involved in the trafficking of α-amino-3-hydroxy-5-methyl-4-isoxazolepropionic acid (AMPA) receptors. The transcriptomics data potentially support the altered metabolism of the glutamine-related pathway.

As glutamine is the main source of amino groups for the production of amino acids, we wondered whether the production of amino acids was affected in the absence of sacsin. Looking at the total levels of amino acids in the undifferentiated cells, there was a small but significant increase in the isoleucine and leucine levels (Figure 4A) and a decrease in aspartate (Figure 4B) in sacsin knockout cells relative to the controls. This latter may have been the result of glucose, and not glutamine-derived carbon cataplerosis, from oxaloacetate in the TCA cycle or occurred independently of the TCA cycle. Considering that aspartate, which showed lower levels relative to the controls (Figure 4B), acts as a substrate for the arginosuccinate synthase reaction, we considered the possibility that it reduced the urea cycle in sacsin knockout cells. Intriguingly, as opposed to the expectation, no difference was present in the levels of urea cycle metabolites (Figure 4C). Differentiated cell lines, contrastingly, did not present any difference in the levels of the amino acids investigated (Figure 4D,E) and presented no difference in urea cycle metabolites (Figure 4F) between the control and sacsin knockout cells.

## 3. Discussion

The work described here indicates that there are bioenergetic alterations in sacsin knockout cells and provides further evidence for mitochondria dysfunction in ARSACS [3,4,5]. Specifically, in the absence of sacsin, cells display decreased levels of glucose and increased levels of lactate, fumarate and malate (Figure 5). However, differentiated cells in the absence of sacsin present increased levels of most TCA cycle metabolites (Figure 5).

In this study, neuronal differentiation had an impact on cell metabolism, which highlights that the data from the cell models may not fully reflect the in vivo situation. Differences between undifferentiated and differentiated SH-SY5Y cells included alterations in the levels of additional TCA cycle metabolites in the absence of sacsin in differentiated cells. It should also be noted that MTA has information on the metabolite levels in the cells at a single time point and that the flux of metabolites is modulated by multiple intrinsic and extrinsic factors [12].

The increased aerobic conversion of glucose to lactate in both undifferentiated and differentiated sacsin knockout cells is reminiscent of alterations in the energy metabolism observed in tumour cells, referred to as the Warburg effect or aerobic glycolysis [13]. In the absence of sacsin, the glucose levels were reduced, as occurs in tumour cells [14]. Interestingly, when lactate undergoes active oxidation, it generates significant amounts of mitochondrial ROS [15]. If not promptly neutralised, excessive ROS can cause oxidative damage that may severely and permanently damage cells [16,17]. This is relevant, as both ARSACS patients’ fibroblasts [3] and sacsin knockout cells [5] display increased ROS levels. Moreover, increased lactate production is a feature of neuronal metabolism in other neurodegenerative disease paradigms. For example, in rodent models of Alzheimer’s disease, astrocytes, microglia and neurons have been found to switch to glycolysis, producing more lactate and exhibiting less TCA cycle activity [18,19,20]. Contrastingly, in the cellular model of ARSACS analysed here, the levels of some TCA cycle metabolites were increased, particularly in the differentiated cell line. The neurobasal media required to culture the differentiated cell lines contained more glucose than the media used for the undifferentiated cells but around half the amount of sodium pyruvate. The different concentrations of the components present in the media might result in a slightly different metabolic activity, potentially translating in the increased levels of all the TCA metabolites evaluated in differentiated sacsin knockout cells relative to the undifferentiated ones.

The metabolic alterations linked to glutamine–glutamate metabolism observed in this study may be relevant to ARSACS. This is because of the neurotransmitter role of glutamate. In the brain, glutamate functions as neurotransmitter, as well as in synapse formation and maturation, is a fuel in oxidative metabolism and has an antioxidant link (as it is the precursor of glutathione) [21]. In ARSACS, Purkinje neurons are highly affected [4,22] and are neurons known to be more susceptible to glutamate-mediated excitotoxicity [23]. Purkinje cells participate in motor control and learning processes by receiving stimulatory input (glutamate-mediated) and releasing inhibitory output (GABA-mediated). Interestingly, in a zebrafish model of ARSACS, the modulation of glutamate neurotransmission by acetyl-DL-leucine, which acts on glutamate release and receptor activation, partially rescued a movement phenotype [24]. Furthermore, in human ataxias of different aetiology, acetyl-DL-leucine has improved the symptoms in most [25,26], although not all, instances [27]. Mitochondrial mislocalisation and altered dynamics are believed to be downstream consequences of the intermediate filament disorganisation observed in ARSACS [22], with related phenotypes as a common feature of other neurodegenerative disorders. Moreover, both high levels of extracellular glutamate and increased ROS levels are features shared with other neurodegenerative diseases, including amyotrophic lateral sclerosis, Alzheimer’s, Parkinson’s, and Huntington’s diseases [28,29].

Mitochondrial dysfunction has been investigated in several other autosomal recessive ataxias, but metabolomics analysis has been limited, with a focus on using untargeted approaches to quantify patient samples (e.g., plasma and urine). This includes a study where both targeted and untargeted metabolomic analysis was used to define a metabolic profile of plasma from Fredrich’s ataxia (FA) patients [30], where deficiency of the protein frataxin leads to dysregulated iron metabolism, the production of reactive oxygen species and decreased bioenergetic efficiency. This work, which was the first reported metabolomic analyses of human patients with FA, identified dysregulated one-carbon metabolism as a key metabolic perturbation [30]. Interestingly, it has been proposed that metabolic imaging approaches may provide a direct methodology to understand the mitochondrial changes occurring in Fredrich’s patients’ brains [31]; although this is technologically challenging, it could be applicable to other ataxias, including ARSACS

One limitation of the study is that we were not able to perform rescue experiments in the knockout cell line, because the large size of sacsin makes its heterologous expression problematic. Another limitation relates to the mass spectrometry technique utilised. This is that less abundant metabolites are sometimes not be detected, either due to matrix effects influencing the ionisation of analytes [32] or that their low level impairs accurate detection. It should also be noted that, ultimately, it will be necessary to investigate the metabolite levels in the cerebellum of sacsin knockout mice to confirm relevance to the in vivo situation and ARSACS’.

In conclusion, this study further supports disruption of the mitochondrial function in the absence of sacsin and provides insights into cellular metabolism, indicating sacsin knockout cells’ reliance on aerobic glycolysis and alteration of the glutamine-related metabolism.

## 4. Materials and Methods

### 4.1. SH-SY5Y Cell Lines, Culture and Differentiation

Human SH-SY5Y neuroblastoma cells were obtained from the American Type Culture Collection and were grown in DMEM/F-12 (1:1) + GlutaMAX (Gibco™ Thermo Fisher Scientific, Loughborough, UK, #31331093), plus 10% heat-inactivated foetal bovine serum (Fisher, #10500-064, and 1% of penicillin–streptomycin (Sigma, Castleford, UK, #P4333-100ML). Cells were incubated at 37 °C with 5% of CO_2_ and grown up to passage 20.

SH-SY5Y cells were grown to near confluence under normal maintenance conditions prior to the start of differentiation, and 1 × 10^5^ cells/well were seeded in a 6-well plate. After 24 h, the media was changed to DMEM low glucose + 1000 mg/L of L-glutamine (Sigma, #D6056) containing 10% foetal bovine serum and 10 µM of retinoic acid (Sigma, #R2625). At day 4, the cells’ media was replaced with neurobasal media (Gibco, #21103049) supplemented with 1× B27 (Gibco, #17504044), 1% GlutaMAX (Thermo Fisher, #35050038), 50 ng/mL brain-derived neurotrophic factor (Sigma, #B3795) and 10 µM of retinoic acid. At day 12, the media was supplemented with 2 mM dibutyryl cyclic AMP (Santa Cruz, Dallas, TX, USA, #sc-201567A). Cells were kept under neurodifferentiating conditions for a further 15 days prior to use in subsequent experiments.

### 4.2. Metabolite Extraction

SH-SY5Y cells were grown to ~80% confluency, then incubated for 16 h in media without the metabolite of interest (either glucose or glutamine), supplemented with 1% penicillin–streptomycin and, if needed, 100 nM sodium pyruvate (Sigma, #S8636), 5% dialysed foetal bovine serum (Sigma, #F0392) and either 17.5 mM of labelled glucose (Sigma, #389374) or 2 mM of labelled glutamine (Cambridge Isotope Laboratories, #CLM-1822-H-0.1, Tewksbury, MA, USA). After 16 h, the metabolites were extracted with extraction buffer (50% methanol, Fisher Scientific, #T001021000, 30% acetonitrile, Fisher Scientific, # A955-1, 20% ultrapure water, Fisher Scientific, # 10434902 and 50 ng/mL HEPES, Sigma, # H4034) at 4 °C. After centrifugation and incubation, samples were transferred into liquid chromatograph autosampler vials (Thermo Fisher, #6PKG592W) and stored at −80 °C until analysis. As control samples, the extraction buffer was used for experimental blanks, an additional control sample was added containing only medium and processed as a regular sample and a pool extract was used mixing 10 µL of each of the samples following the extraction. A minimum of three biological replicates were used for the analysis. Fold change > 2 and *p* < 0.05 were used as the threshold. LC-MS/MS analysis of the metabolites was performed using a Q Exactive Quadrupole-Orbitrap mass spectrometer attached to a Vanquish UHPLC system (Thermo Fisher Scientific).

### 4.3. Confluency Assay

Cells were split in a 6-well plate. Each well was then treated with either regular medium or glucose-free media or vehicle-treated or sodium oxamate (50 nM, Cambridge Bioscience, #CAY19057, Cambridge, UK). Cells were imaged and percentage confluence was measured using an Incucyte SX5 live cell imaging analyser.

### 4.4. Immunoblot

Cells were lysed with RIPA buffer (Sigma, #20188) and prepared with 1:1 volume of 2× Laemmli buffer (Merck, #S3401, Darmstadt, Germany), and 10 ug of protein was loaded per lane on a Novex 4–12% tris-glycine gel (Invitrogen, #NP0335/6BOX, Waltham, MA, USA). The protein was transferred to PVDF (VWR, #10600002, Radnor, PA, USA), and for the sacsin blot, an overnight wet transfer was performed. The membrane was then blocked with 1× bovine serum albumin (Merck, #A7906). Blots were washed and incubated with primary antibodies at 1:1000 dilution β-III Tubulin (ab18207), β-actin (ab8226), sacsin (ab181190) and OXPHOS (ab110413), followed by HRP secondary antibodies at 1:1000 dilution IRDye680 (P/N: 926-68070) and IRDye800 CW (P/N: 926-32211) (LiCor, Lincoln, NE, USA). The protein was quantified using ImageJ software (version 1.53). Each lane was normalised to the relative density of β-actin or total protein with Fast green.

### 4.5. Statistical Analysis

All statistical analyses were performed using GraphPad Prism version 10.3.1. All data were expressed as the mean ± SD, with a minimum of three biological replicates used for each analysis. Statistical significances (*p*-value) are presented in the graph for the values <0.1. For all the metabolomic tracer analyses, the raw value for the peak areas of the metabolites of interest was normalised to the total ion count (TIC), which was the sum of every metabolite found in the samples. The significance level was 0.05 for all tests. Statistical differences between 2 groups were determined using an unpaired *t*-test. Isotopomers distribution analysis was analysed using ordinary one-way ANOVA with Tukey’s multiple comparison test. Cell confluency was analysed with two-way ANOVA with Tukey’s multiple comparison test.

## Figures and Tables

**Figure 1 ijms-25-13242-f001:**
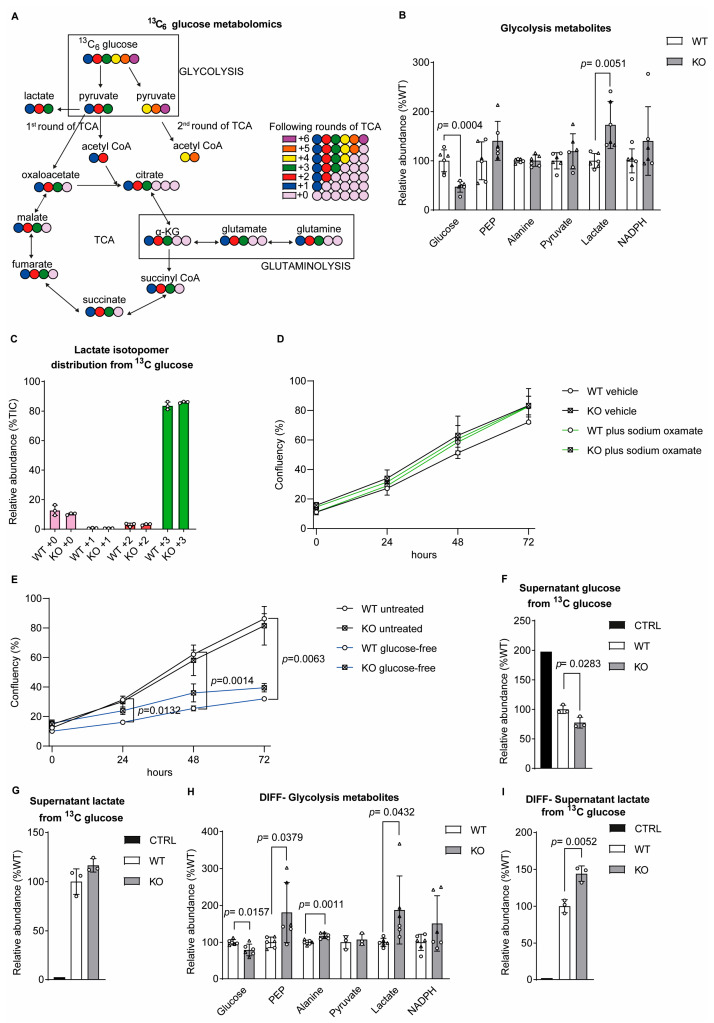
Sacsin knockout cells exhibit increased lactate production. (**A**) Schematic representation of [U-13C] glucose tracer labelling of cellular metabolites. The six carbons of glucose are represented in different colours, and in each round of glycolysis, three of these carbons are converted into pyruvate and then incorporated into other metabolites. (**B**) Levels of glycolysis metabolites from both glucose and glutamine labelling in SH-SY5Y cells. (**C**) Percentage of lactate isotopomers distribution from [U-13C]-traced glucose of the cells. (**D**,**E**) Cell confluency was assessed for both wild-type control and sacsin knockout cells in vehicle-treated media and in media supplemented with sodium oxamate (**D**), in untreated media and glucose-free media (**E**) over 3 days. Cell confluency quantification was performed with Incucyte S3 software analysis. Two-way ANOVA with Tukey’s multiple comparison test was applied, *n* = 3. (**F**,**G**) Levels of (**F**) glucose and (**G**) lactate in cell culture media collected after 16 h. An additional control sample was added containing only the medium and processed as a regular sample. (**H**) Levels of glycolysis metabolites from both glucose and glutamine labelling in differentiated SH-SY5Y cells. (**I**) Levels of lactate in differentiated spent medium of SH-SY5Y cells. Levels of each isotopomer are expressed as a percentage of the total. The significance level was 0.05 (test used Anderson–Darling). The raw value, for the peak areas of the metabolites of interest, has been normalised to the total ion count (TIC), which is the sum of every metabolite found in the samples. Total metabolites (**B**–**I**) unpaired *t*-test, isotopomers distribution (**C**) one-way ANOVA. All error bars are S.D. ○ = traced glucose, ∆ = traced glutamine, KG = ketoglutarate, TCA = tricarboxylic acid cycle, CoA = coenzyme A, PEP = phosphoenolpyruvate, WT = wild-type control, KO = sacsin knockout, CTRL = media control, Undiff = undifferentiated SH-SY5Y and diff = differentiated SH-SY5Y.

**Figure 2 ijms-25-13242-f002:**
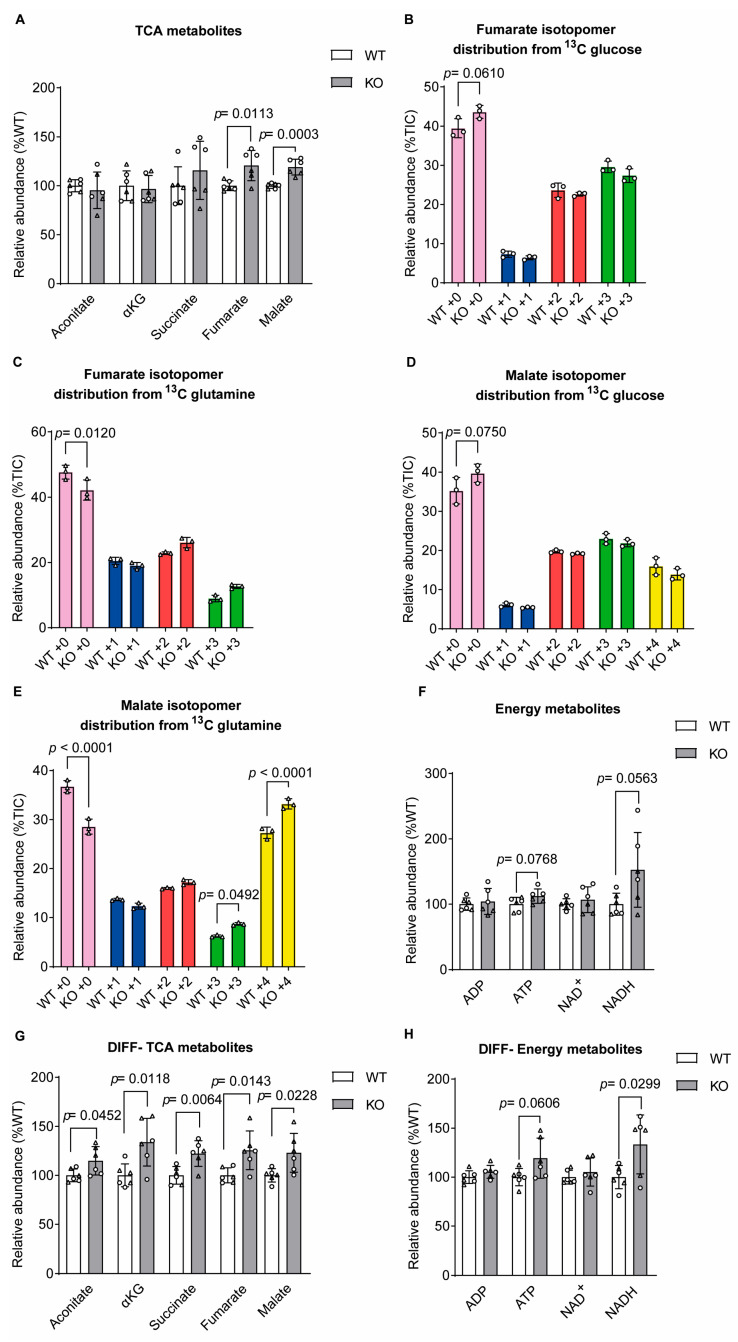
Malate levels were increased in sacsin knockout cells. (**A**) Levels of TCA metabolites in wild-type control and sacsin knockout SH-SY5Y cells from both glucose and glutamine labelling. (**B**,**D**) Distribution of fumarate (**B**) and malate (**D**) isotopomers arising from [U-13C]-traced glucose after labelling of the control and sacsin knockout cells. (**C**,**E**) Distribution of fumarate (**C**) and malate (**E**) isotopomers arising from [U-13C]-traced glutamine after labelling of the control and sacsin knockout cells. (**F**) Total levels of ADP, ATP, NAD^+^ and NADH in the control and sacsin knockout cells from both glucose and glutamine labelling. (**G**) Levels of TCA cycle metabolites from both glucose and glutamine labelling in differentiated SH-SY5Y cells. (**H**) Total levels of ADP, ATP, NAD^+^ and NADH in differentiated control and sacsin knockout cells from both glucose and glutamine labelling. Levels of each isotopomer are expressed as a percentage of the total. Data are normalised to the total ion count (TIC). Total metabolites (**A**,**F**–**H**) unpaired *t*-test, isotopomers distribution (**B**–**E**) one-way ANOVA. All error bars are S.D. ○ = traced glucose, ∆ = traced glutamine, KG = ketoglutarate, TCA = tricarboxylic acid cycle, WT = wild-type control, KO = sacsin knockout, Undiff = undifferentiated SH-SY5Y and diff = differentiated SH-SY5Y.

**Figure 3 ijms-25-13242-f003:**
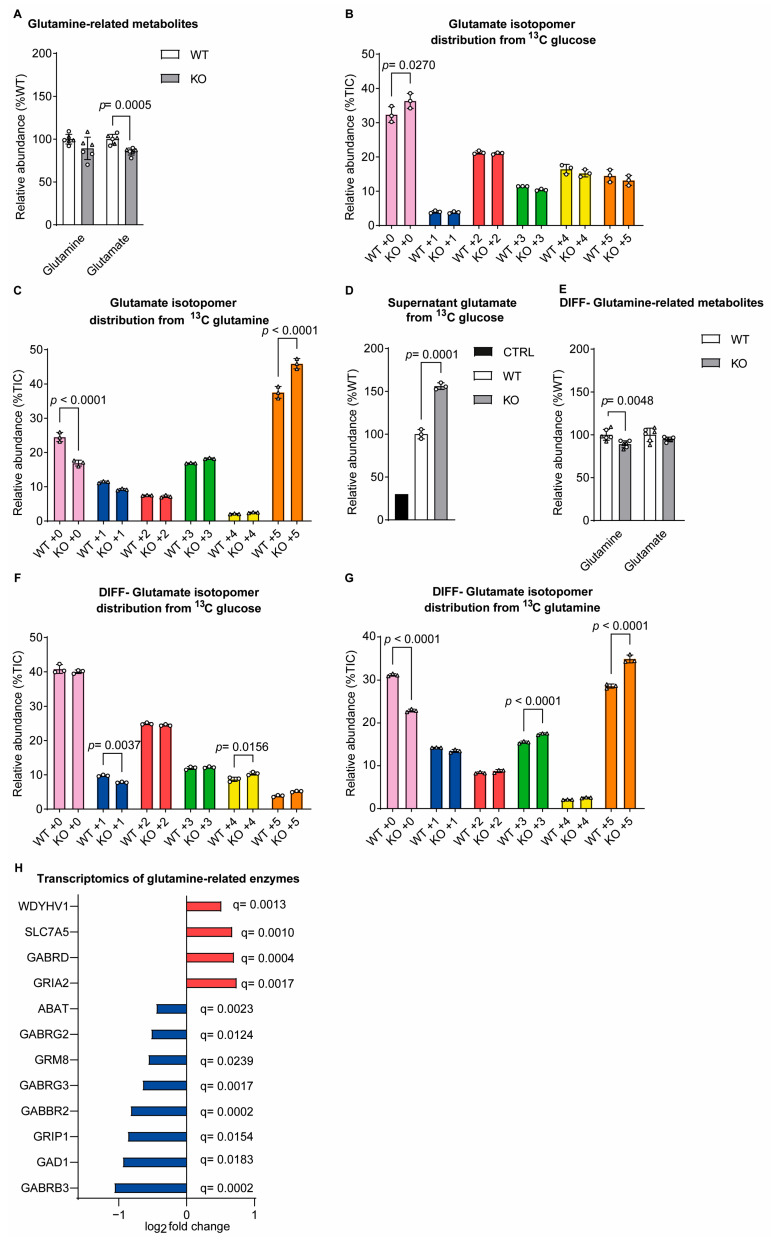
Glutamate levels were decreased in sacsin knockout cells. (**A**) Levels of glutamine-related metabolites from both glucose and glutamine labelling in SH-SY5Y cells. (**B**,**C**) Distribution of glutamate isotopomers arising from (**B**) [U-13C]-traced glucose or (**C**) [U-13C]-traced glutamine after labelling of control and sacsin knockout cells. (**D**) Levels of glutamate in spent medium of [U-13C]-traced glucose. (**E**) Levels of glutamine-related metabolites from both glucose and glutamine labelling in differentiated SH-SY5Y cells. (**F**,**G**) Distribution of glutamate isotopomers arising from (**F**) [U-13C]-traced glucose or (**G**) [U-13C]-traced glutamine after labelling of control and sacsin knockout cells. (**H**) Levels of transcript for enzymes involved in the glutamine-related pathway in sacsin knockout normalised to the controls. Data are for differentially expressed genes (*p* < 0.05), from the transcriptomics analysis of differentiated SH-SY5Y cells [8]. False discovery rate q values are displayed on the graph. Data are normalised to the total ion count (TIC). total metabolites unpaired *t*-test. All error bars are the S.D. ○ = traced glucose, ∆ = traced glutamine, WT = wild-type control, KO = sacsin knockout, CTRL = media control, Undiff = undifferentiated SH-SY5Y and diff = differentiated SH-SY5Y.

**Figure 4 ijms-25-13242-f004:**
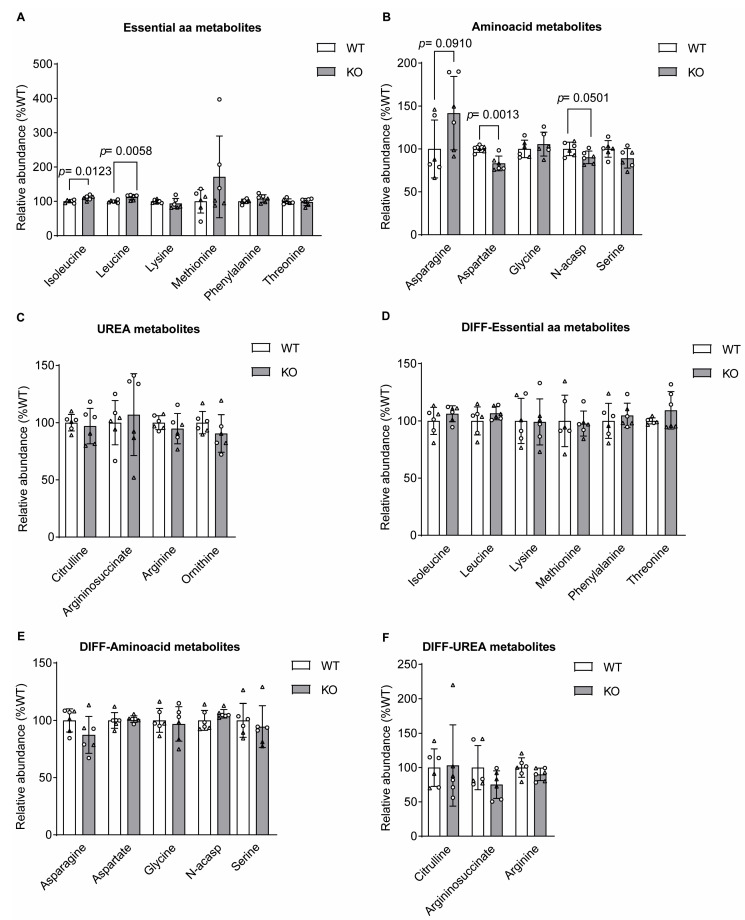
Isoleucine and leucine levels were increased in the absence of sacsin. (**A**–**C**) Levels of (**A**) essential amino acids, (**B**) amino acids and (**C**) urea cycle metabolites from both glucose and glutamine labelling in SH-SY5Y cells. (**D**–**F**) Levels of (**D**) essential amino acids, (**E**) amino acids and (**F**) urea cycle metabolites from both glucose and glutamine labelling in differentiated SH-SY5Y cells. Data are normalised to the total ion count (TIC). total metabolites unpaired *t*-test. All error bars are the S.D. ○ = traced glucose, ∆ = traced glutamine, diff = differentiated SH-SY5Y, WT = wild-type control, KO = sacsin knockout and N-acasp = n-acetylaspartate.

**Figure 5 ijms-25-13242-f005:**
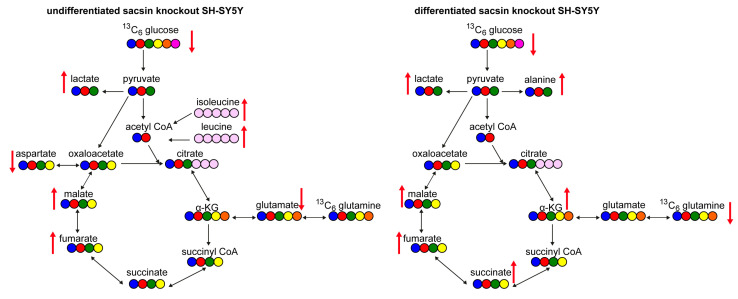
Summary of the metabolic alteration in the absence of sacsin. Schematic representation of altered metabolites in the absence of sacsin on the left in undifferentiated cell lines, and on the right in differentiated cell lines, using both [U-13C] glucose and [U-13C] glutamine tracer labelling. Red arrows indicate if levels of metabolites are increased or decreased. Black arrows show metabolic pathways.

## Data Availability

The data presented in this study are available on request from the corresponding author.

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
