# Peer review of "Altered Cellular Metabolism Is a Consequence of Loss of the Ataxia-Linked Protein Sacsin"

_ijms, 2024, doi:10.3390/ijms252413242_

Round 1

Reviewer 1 Report

Comments and Suggestions for Authors

Perna et al have found out that sacsin-knockout SH-SY5Y neuroblastoma cells exhibit the enhanced aerobic glycolysis accompanied by increase in lactate and reduction of glucose. The authors also observed disruption in the glutaminolysis pathway in differentiated and undifferentiated cells in the absence of sacsin. These alterations in the metabolism of both glucose and glutamate may provide an important basis for the mitochondrial dysfunction in Autosomal Recessive Spastic Ataxia of Charlevoix Saguenay (ARSACS). Overall this manuscript is well written; however, there are a couple of concerns:

1.       I’d think that rescue experiments are necessary: for example, does transfection of sacsin cDNA suppress the enhanced aerobic glycolysis down to the control level?

2.       Do these metabolic changes really happen in the cerebellum?

Author Response

Perna et al have found out that sacsin-knockout SH-SY5Y neuroblastoma cells exhibit the enhanced aerobic glycolysis accompanied by increase in lactate and reduction of glucose. The authors also observed disruption in the glutaminolysis pathway in differentiated and undifferentiated cells in the absence of sacsin. These alterations in the metabolism of both glucose and glutamate may provide an important basis for the mitochondrial dysfunction in Autosomal Recessive Spastic Ataxia of Charlevoix Saguenay (ARSACS). Overall this manuscript is well written; however, there are a couple of concerns:

  1. I’d think that rescue experiments are necessary: for example, does transfection of sacsin cDNA suppress the enhanced aerobic glycolysis down to the control level?

The rescue experiment proposed by the reviewer would be useful but has proved extremely technically challenging. This is because sacsin is a massive protein (4578 aa, 520 KDa) where the full-length open reading frame has not been amenable to cloning into routinely used expression vectors, or the protein to heterologous expression. A sacsin-GFP construct is available, but transfection efficiency is extremely low and heterologous expressed protein is not detectable by Western blot (although a small incidence of cells expressing low levels of sacsin-GFP were detected by confocal). Some imaging-based studies have looked at isolated domains of sacsin (e.g., J-domain, SIRPT domains) but none have reported an influence on mitochondrial phenotypes. We have tested similar sacsin domain expression vectors and again observe low numbers of cells expressing detectable levels of the heterologous expressed proteins. As the tracer analysis measures metabolites in cell lysates, we believe robust expression would be necessary to get meaningful results from rescue experiments. This limitation of the work is now referred to in the discussion - ‘One limitation of the study is that we were not able to perform rescue experiments in the knockout cell line because the large size of sacsin makes its heterologous expression problematic’.

  1. Do these metabolic changes really happen in the cerebellum?

This is an excellent question but undertaking in vivo metabolic tracer analysis is highly demanding requiring highly specialised expertise and instrumentation. Moreover, hardly any researchers appear to have performed in vivo tracer analysis in the murine cerebellum, and we are not aware of any studies in models of neurodevelopmental or neurodegenerative disease. We thus believe such work is beyond the scope of the current study. This limitation of the work is now referred to in the discussion -‘..ultimately, it will be necessary to investigate metabolite levels in the cerebellum of sacsin knockout mice to confirm relevance to the in vivo situation and ARSACS’.

Reviewer 2 Report

Comments and Suggestions for Authors

In general, the study is confused, but I think the main problem is in Methods section. Some Methods information is described in results section, which is difficult to understand. Besides of that, absence of some important points in methods section makes impossibles study replicability.

Introduction

Please clarify study objective and hypothesis.

Add Methods section, describing methodology used in the study.

Results

Line 127 Add which normality test used.
line 128, lune 164, 232: Better explain variables analyzed by unpaired t-test and which was analyzed by one-way ANOVA

Line 129-132 and line 166 to 168, line 233 and 234 This information should be added in figure legend.

Line 142 Which statistic tests was used in these results?

Put together graphics makes them confuse to understand.

Discussion

Add study limitation, clinical implication and suggestions for future studies.

Methods

Add information about statistic analysis.

Add a study’s flowchart, this can help readers to understand the study.  

Author Response

In general, the study is confused, but I think the main problem is in Methods section. Some Methods information is described in results section, which is difficult to understand. Besides of that, absence of some important points in methods section makes impossibles study replicability.

We have responded to all of the reviewer’s specific points and improved the method section of the paper as described below.

Introduction - Please clarify study objective and hypothesis.

We have edited the final paragraph of the introduction to more clearly emphasise the study objective and hypothesis to be tested. This text now reads - ‘The objective of this study was to better understand the mitochondrial deficit in ARSACS and specifically test the hypothesis that mitochondrial metabolism would be disrupted. To achieve this goal, we performed metabolic tracer analysis (MTA) of both 13C isotopic labelled glucose and glutamine by mass spectrometry.’

Add Methods section, describing methodology used in the study. 

We improved the methods section adding the statistical analysis section and the study flowchart as requested by reviewer.

Results

Line 127 Add which normality test used. 

The reviewer is referring to the sentence ‘Data are normalised to the total ion count (TIC)’. The α level was 0.05 (test used Anderson-Darling). The raw value, for the peak areas of the metabolites of interest, has been normalised to the total ion count (TIC), which is the sum of every metabolite found in the samples. This sentence has been now added to the statistical analysis section in the material and methods.

line 128, lune 164, 232: Better explain variables analyzed by unpaired t-test and which was analyzed by one-way ANOVA

The reviewer is referring to the sentence ‘Total metabolites unpaired t-test, isotopomers distribution one-way ANOVA.’ Now the sentence under each figure legend will be changed using the following scheme: Total metabolites (B,F,G,H,I) unpaired t-test, isotopomers distribution (C) one-way ANOVA.

Line 129-132 and line 166 to 168, line 233 and 234 This information should be added in figure legend. 

We do not understand the reviewer’s comment as this information is already in the figure legends. Please could they clarify.

Line 142 Which statistic tests was used in these results?

The reviewer is referring to the sentence ‘An increase, although not statistically significant, can be seen in the total ATP (p-value 0.0768) and NADH (p-value 0.0563), but not in ADP and NAD+ (Figure 2F) levels in sacsin knockout cells relative to controls.’ The figure legend will now include Total metabolites‘ (A,F,G,H) unpaired t-test, isotopomers distribution (B,C,D,E) one-way ANOVA’.

Put together graphics makes them confuse to understand.

We do not understand the reviewer’s comment. Please could they clarify which figure panel they are unhappy with and why.

Discussion - Add study limitation, clinical implication and suggestions for future studies. 

We have now discussed some limitations. Including responding to the comments made by reviewer 1 regarding rescue experiments and understanding whether metabolic changes are recapitulated in cerebellum. We also added the following sentence ‘Another limitation relates to the mass-spectrometry technique utilized. This is that less abundant metabolites are sometimes not be detected, either due to matrix effects influencing the ionisation of analytes (Alseekh et al., 2021) or that their low level impairs accurate detection.’

Methods

Add information about statistic analysis. 

A statistical analysis section has been added with the following information:

All statistical analysis were performed using GraphPad Prism version 10.3.1. All data are expressed as the mean ± SD with a minimum of three biological replicates used for each analysis. Statistical significances (p-value) are presented in the graph for the values <0.1. For all the metabolomic tracer analysis, the raw value for the peak areas of the metabolites of interest has been normalised to the total ion count (TIC), which is the sum of every metabolite found in the samples. The significance level was 0.05 for all tests, Shapiro-Wilk, Kolmogorov-Smirnov). Statistical differences between 2 groups were determined using unpaired t-test. Isotopomers distribution analysis was analysed using ordinary one-way ANOVA with Tukey’s multiple comparison test. Cell confluency was analysed with two-way ANOVA with Tukey’s multiple comparison test.

Add a study’s flowchart, this can help readers to understand the study.  

A flow chart has been inserted as part of Figure S1.

Reviewer 3 Report

Comments and Suggestions for Authors

The paper is adequately designed. It could be interesting that you compare the mitochondrial disfunction in ARSACS to one in FRDA and/or CANVAS and other ataxias, at least as a brief literature review.

Comments on the Quality of English Language

The quality of English can be easily improved to more clearly explain your work, which is really interesting. 

Line 69: you can correct were...perhaps where glucose?

Line 95: you can correct more neuronal?

Line 236: indicates there.....You can add indicates that there...

Author Response

The paper is adequately designed. It could be interesting that you compare the mitochondrial disfunction in ARSACS to one in FRDA and/or CANVAS and other ataxias, at least as a brief literature review.

The following discussion has been included – ‘Mitochondrial dysfunction has been investigated in several other autosomal recessive ataxias, but metabolomics analysis has been limited, with a focus on using untargeted approaches to quantify patient samples (e.g., plasma, urine). This includes a study where both targeted and untargeted metabolomic analysis was used to define a metabolic profile of plasma from Fredrich’s ataxia (FA) patients (O'Connell et al., 2022), where deficiency of the protein frataxin leads to dysregulated iron metabolism, production of reactive oxygen species and decreased bioenergetic efficiency. This work, which was the first reported metabolomic analyses of human patients with FA, identified dysregulated one-carbon metabolism as a key metabolic perturbation (O'Connell et al., 2022). Interestingly, it has been proposed that metabolic imaging approaches may provide a direct methodology to understand the mitochondrial changes occurring in Fredrich’s patients brain (Lynch and Farmer, 2021), although this is technologically challenging, it could be applicable to other ataxias including ARSACS’.

Comments on the Quality of English Language

The quality of English can be easily improved to more clearly explain your work, which is really interesting. 

Line 69: you can correct were...perhaps where glucose?

The reviewer is referring to the sentence ‘For 13C glucose labelling PC1 (which accounted for 93.99% of the total variance) separated the controls from all knockout lines (Figure S1B), while for 13C glutamine labelling they were separated by PC2 (which accounted for 28.33% of the total variance)’. In this sentence ‘were’ is correct but we have edited it to read ‘For 13C glucose labelling, PC1 (which accounted for 93.99% of the total variance) separated the controls from knockouts (Figure S1B), while for 13C glutamine labelling, the controls and knockout cells were distinguished by PC2 (which accounted for 28.33% of the total variance)’

Line 95: you can correct more neuronal?

The sentence ‘Given the importance of glutamate and related metabolites in neurotransmission and relevance to neurological diseases, we next differentiated the isogenic control and sacsin knockout cells to a more neuronal phenotype’ has been changed to ‘Given the importance of glutamate and related metabolites in neurotransmission and their relevance to neurological diseases, we next undertook neuronal differentiation of the isogenic control and sacsin knockout cells’.

Line 236: indicates there.....You can add indicates that there...

This edit has been made such that the sentence now reads – ‘The work described here indicates that there are bioenergetic alterations in sacsin knockout cells and provides further evidence for mitochondria dysfunction in ARSACS [3-5]. Specifically, in the absence of sacsin cells display decreased levels of glucose, and increased levels of lactate, fumarate and malate.’

Round 2

Reviewer 1 Report

Comments and Suggestions for Authors

I have nothing to comment any more.

Reviewer 2 Report

Comments and Suggestions for Authors

I thank the authors for the answers and modifications based on previous suggestions.

Regarding the previous comment "Line 129-132 and line 166 to 168, line 233 and 234 This information should be added in figure legend. " I apologize for not being clear, I believe that this information should only come in the figure legend and not in the main text.

Beyond that I have no further comments.